# An Exploration of Pathologies of Multilevel Principal Components Analysis in Statistical Models of Shape

**DOI:** 10.3390/jimaging8030063

**Published:** 2022-03-04

**Authors:** Damian J. J. Farnell

**Affiliations:** School of Dentistry, Cardiff University, Heath Park, Cardiff CF14 4XY, UK; farnelld@cardiff.ac.uk

**Keywords:** multilevel principal components analysis (mPCA), 3D shape analysis, Monte Carlo simulations

## Abstract

3D facial surface imaging is a useful tool in dentistry and in terms of diagnostics and treatment planning. Between-group PCA (bgPCA) is a method that has been used to analyse shapes in biological morphometrics, although various “pathologies” of bgPCA have recently been proposed. Monte Carlo (MC) simulated datasets were created here in order to explore “pathologies” of multilevel PCA (mPCA), where mPCA with two levels is equivalent to bgPCA. The first set of MC experiments involved 300 uncorrelated normally distributed variables, whereas the second set of MC experiments used correlated multivariate MC data describing 3D facial shape. We confirmed results of numerical experiments from other researchers that indicated that bgPCA (and so also mPCA) can give a false impression of strong differences in component scores between groups when there is none in reality. These spurious differences in component scores via mPCA decreased significantly as the sample sizes per group were increased. Eigenvalues via mPCA were also found to be strongly affected by imbalances in sample sizes per group, although this problem was removed by using weighted forms of covariance matrices suggested by the maximum likelihood solution of the two-level model. However, this did not solve problems of spurious differences between groups in these simulations, which was driven by very small sample sizes in one group. As a “rule of thumb” only, all of our experiments indicate that reasonable results are obtained when sample sizes per group in all groups are at least equal to the number of variables. Interestingly, the sum of all eigenvalues over both levels via mPCA scaled approximately linearly with the inverse of the sample size per group in all experiments. Finally, between-group variation was added explicitly to the MC data generation model in two experiments considered here. Results for the sum of all eigenvalues via mPCA predicted the asymptotic amount for the total amount of variance correctly in this case, whereas standard “single-level” PCA underestimated this quantity.

## 1. Introduction

Geometric morphometrics aims to provide a mathematical description of biological shapes [1,2,3,4,5]. Three-dimensional (3D) surface scanning [6] is a technique that allows one to capture the 3D shape, e.g., of the human face, as shown in Figure 1 for the author’s face. It is a useful tool in understanding dental and maxillofacial diagnostics, treatment planning, and the effects of treatment [7]. Such biological shapes may be described by a set of landmark points, illustrated also in Figure 1. Methods such as Procrustes transformation [1] are often used to standardise centring, orientation, and scale in a dataset of such 3D shapes.

Multivariate data contains more than one “outcome” variable, such as the *x*-, *y*- and *z*-components of the Cartesian landmark points (again, as shown in Figure 1). These variables tend to be highly correlated and so multivariate statistical methods such as principal components analysis (PCA) [1] are needed to in order to analyse such data. Between-group PCA (bgPCA) [8,9] is an extension of standard PCA that carries out separate PCAs on (between-group) covariance matrices based on “group means” and (within group) covariance matrices based on individual shapes around these means. It has much in common with (though it is not the same as) canonical variates analysis/linear discriminant analysis [10]. Multilevel PCA (mPCA) has been used by us [11,12,13,14,15,16,17] to analyse 3D facial shapes obtained from 3D facial scans; note that two-level multilevel PCA (mPCA) is equivalent to bgPCA. mPCA has been used by us to investigate changes by ethnicity and sex [11,12], the act of smiling [13,14], facial shape changes in adolescents due to age [15,16], and the effects of maternal smoking and alcohol consumption on the facial shape of English adolescents [17].

Recent articles [8,9,18,19,20] have pointed out a number of “pathologies” in techniques such as bgPCA (and therefore also mPCA). Perhaps the most notable pathology [8,9] is that spurious conclusions about differences between groups can occur when the number of parameters in the model is much larger than sample sizes used in the model. Another limitation occurs when sample sizes are not balanced between groups [8]. Here, we wish to explore these pathologies by carrying out Monte Carlo simulated experiments firstly using uncorrelated normally distributed variables and secondly for correlated multivariate normally distributed data based on “real” data using the 21 landmark points in Figure 1.

## 2. Materials and Methods

### 2.1. Monte Carlo Data Generation

Data is generated via Monte Carlo techniques and a number of “experiments” are carried out here, as shown in Table 1. For Experiment 1, p=300 uncorrelated standard normal distribution (i.e., mean = 0 and standard deviation = 1) are used, exactly as in [8]. Data for each variable is generated using the randn() command in MATLAB (R2021a). The number of groups is set to be equal to 3 in the results presented here, and the sample size per group $n_{l}$ is varied explicitly, where l indicates a specific group. The overall sample size for m groups is given by $n=\overset{m}{\underset{l}{\sum }}n_{l}$. Note that the limits are placed on the magnitude of eigenvalues via PCA by the Marchenko–Pastur theorem [21] for such random (uncorrelated) data such that eigenvalues must lie between ${\left(\sqrt{y}-1\right)}^{2}$ and ${\left(\sqrt{y}+1\right)}^{2}$, where y=p/n in the limit that both p and n tend to infinity. For Experiment 2, 300 uncorrelated standard variables are again used in these simulations. In order to explore imbalances in sample sizes, the sample size per group in group 3 is set to be *n* = 10, whereas the sample size per group $n_{1,2}$ for the two other groups is varied explicitly. There is no between-group variation in Experiments 1 and 2, and so the total variance over all 300 independent/uncorrelated variables is also equal to 300. For Experiment 3, 300 uncorrelated standard normally distributed variables are used at level 2 of the model in Figure 2, although between-group variation is allowed in this case explicitly. A constant offset for each group of subjects is added to all variables, where this offset itself follows a normal distribution with mean = 0 and standard deviation = 0.25 (here) and is independent of the “within group” source of variations. For each variable, between-group variance equals 0.25^2^ = 0.0625 and “within-group” variance equals 1. The total variance over all 300 mutually independent variables is equal to: 300 × 1.0625 = 318.75. The percentage of the total variance explained due by between-group variation (Equation (3) below) in Experiment 3 is given by 5.7% (= 100 × 18.75/318.75), whereas this percentage is clearly 0% in Experiments 1 and 2. 100 MC datasets are used in these simulations for Experiments 1 to 3, except for when the sample size per group was equal to 300 (where only 50 MC datasets were used due to computational demands).

Experiments 4 and 5 use data from [12] for 21 3D landmark points (i.e., 21 × 3 = 63 variables), again as shown in Figure 1. This landmark point data was transformed using a generalised Procrustes analysis [1]. A Procrustes transformation is one that involves translation, rotation, or uniform scaling (or a combination of all of them). Here, it was used to form a common origin, orientation and length scale for all shapes represented by the sets of landmark points with respect to a reference given by the mean shape. This data is used here to form an average covariance matrix (over the two matrices for males and females separately), which is then used in a multivariate normal random number generator (i.e., the mvnrnd() command in MATLAB) in order to create the MC data. Variables are therefore correlated in Experiments 4 and 5. Only two groups of equal sample size ($n_{1}=n_{2}$) that correspond to males and females in the original dataset are employed in Experiments 4 and 5. However, Experiment 4 uses a single mean vector (i.e., an average over subjects for the data in [12]), thereby implying that there is no between-group variation for Experiment 4. Any differences between “males” and “females” in the two groups are therefore spurious in this case and the percentage of total variance due to between-group variation (Equation (3) below) is equal to zero asymptotically with respect to increasing sample size. By contrast, Experiment 5 assumes separate mean shape vectors for males and females (again obtained directly from data in [12]), which implies that between-group variance at level 1 is non-zero in this case. Note that we find a value of 10.4% for the percentage of total variance due to between-group variation (Equation (3) below) directly from the original experimental data ($n_{1}=124;\;n_{2}=126$) in this case. A total of 100 MC datasets are used in all simulations for Experiments 4 and 5.

### 2.2. Multilevel Principal Components Analysis (mPCA)

Features (i.e., 300 variables here for Experiments 1 to 3 and 63 components for Experiments 4 and 5) are represented by a vector z. Single-level PCA is carried out by finding the mean vector μ over all data points and a covariance matrix given by
(1)$$\Sigma_{k_{1},k_{2}}=\frac{1}{n-1}\sum_{i=1}^{n}\left(z_{ik_{1}}-{\overset{-}{\mu }}_{k_{1}}\right)\left(z_{ik_{2}}-{\overset{-}{\mu }}_{k_{2}}\right).$$

$k_{1}$ and $k_{2}$ indicate elements of this covariance matrix and i refers to a given data point in the set. The eigenvalues $\lambda_{l}$ and (orthonormal) eigenvectors $u_{l}$ of this matrix are found readily. Note that the rank of this covariance matrix (and so also the number of non-zero eigenvalues) is limited to n−1. For PCA, one ranks all the eigenvalues into descending order, and one retains the first $p_{1}$ components in the model. The vector z is modeled by
(2)$$z^{\mathrm{PCA}}=\mu +\sum_{l=1}^{p_{1}}a_{l}u_{l}.$$

The coefficients $\left\{a_{l}\right\}$ (also referred to here as “component scores”) are found readily by using a scalar product with respect to the set of orthonormal eigenvectors, i.e., $a_{l}=u_{l}\cdot \left(z-\mu \right)$, for a fit of the model to a new vector z.

The mPCA model used here is illustrated schematically in Figure 2. Note that separate covariance matrices are found at levels 1 and 2 for mPCA. Group means at level 2 are denoted $\mu_{l}^{2}$ and the covariance matrix $\Sigma^{2}$ at level 2 is just the average of all of the “local” covariance matrices $\Sigma_{l}^{2}$ for each group l. (The rank of each of these covariance matrices is limited to $n_{l}-1$). The overall “grand mean” at level 1 (denoted by $\mu^{1}$) is the average over all local group means $\mu_{l}^{2}$ at level 2, i.e., $\mu^{1}=\sum_{l=1}^{m}\mu_{l}^{2}/m$. The level 1 covariance matrix is given by $\Sigma^{1}=\sum_{l=1}^{m}{\left(\mu_{l}^{2}-\mu^{1}\right)}^{2}/\left(m-1\right)$, where m is the number of groups. (The rank of this covariance matrix is limited to m−1). Both of these covariance matrices are diagonalised separately, where each eigenvalue at level 1 is denoted by $\lambda_{l}^{1}$, with associated eigenvector $u_{l}^{1}$, and each eigenvalue at level 2 is denoted by $\lambda_{l}^{2}$, with associated eigenvector $u_{l}^{2}$. We rank the eigenvalues into descending order at each level of the model separately, and then we retain the first $p_{1}$ and $p_{2}$ eigenvectors of largest magnitude at the two levels. The percentage variation at level 1 via mPCA with respect to the overall variation is
(3)$$\mathrm{Percentage}\;\mathrm{Variation}\;\mathrm{At}\;\mathrm{Level}\;1=100\times \frac{\sum_{l=1}^{p_{1}}\lambda_{l}^{1}}{\sum_{l=1}^{p_{1}}\lambda_{l}^{1}+\sum_{l=1}^{p_{2}}\lambda_{l}^{2}}.$$

The vector z is modeled for the two-level model shown in Figure 2 by
(4)$$z^{\mathrm{mPCA}}=\mu^{1}+\sum_{l=1}^{p_{1}}a_{l}^{1}u_{l}^{1}+\sum_{l=1}^{p_{2}}a_{l}^{2}u_{l}^{2}.$$

The coefficients $\left\{a_{l}^{1}\right\}$ and $\left\{a_{l}^{2}\right\}$ (also referred to as “component scores” here) are determined for mPCA by using a global optimisation procedure in MATLAB.

### 2.3. Maximum Likelihood Solution

In order to find the maximum likelihood solution, we assume an expression for the likelihood function of a two-level model that is given by
(5)$$L=\prod_{l=1}^{m}\prod_{i=1}^{n_{l}}N(\mu_{l}^{2}|\mu^{1}\Sigma^{1})N(z_{i}|\mu_{l}^{2}\Sigma^{2}).$$

$N(z_{i}|\mu_{l}^{2}\Sigma^{2})$ is a multivariate normal distribution, where $\mu_{l}^{2}$ is the mean for group l and $\Sigma^{2}$ is the covariance matrix at level 2 (assuming here a common covariance matrix at this level as for mPCA). $n_{l}$ is sample size for group l and m is the number of groups. $N(\mu_{l}^{2}|\mu^{1}\Sigma^{1})$ is another multivariate normal distribution, where $\mu^{1}$ is the “grand” mean and $\Sigma^{1}$ is the covariance matrix at level 1. The associated log likelihood (LL) is
(6)$$LL\propto \frac{n}{2}\left(\ln\left|\Sigma^{1}\right|+\ln\left|\Sigma^{2}\right|\right)-\frac{1}{2}\sum_{l=1}^{m}n_{l}\left({\left(\mu_{l}^{2}-\mu^{1}\right)}^{T}{\left(\Sigma^{1}\right)}^{-1}(\mu_{l}^{2}-\mu^{1})\right)-\frac{1}{2}\sum_{l=1}^{m}\sum_{i=1}^{n_{l}}\left({\left(z_{i}-\mu_{l}^{2}\right)}^{T}{\left(\Sigma^{2}\right)}^{-1}(z_{i}-\mu_{l}^{2})\right).$$

The maximum likelihood solution is therefore given by
(7)$$\begin{array}{cc} \frac{\partial LL}{\partial \mu^{1}}=0\Rightarrow \mu^{1}=\frac{1}{n}\sum_{l=1}^{m}n_{l}\mu_{l}^{2}(=\mu ) & \frac{\partial LL}{\partial {\left(\Sigma^{1}\right)}^{-1}}=0\Rightarrow \Sigma^{1}=\frac{1}{n}\sum_{l=1}^{m}n_{l}(\mu_{l}^{2}-\mu^{1}){\left(\mu_{l}^{2}-\mu^{1}\right)}^{T} \end{array},$$
and
(8)$$\begin{array}{cc} \frac{\partial LL}{\partial \mu_{l}^{2}}=0\Rightarrow \mu_{l}^{2}=\frac{1}{n_{l}}\sum_{i=1}^{n_{l}}z_{i} & \frac{\partial LL}{\partial {\left(\Sigma^{2}\right)}^{-1}}=0\Rightarrow \Sigma^{2}=\frac{1}{n}\sum_{l=1}^{m}\sum_{i=1}^{n_{l}}(z_{i}-\mu_{l}^{2}){\left(z_{i}-\mu_{l}^{2}\right)}^{T} \end{array}.$$

Equations (7) and (8) are (almost) identical to the equations used in mPCA presented above when sample sizes per group $n_{l}$ are equal to each other for all groups l, although they are “population” rather than “sample” covariance matrices in this case (i.e., there is a factor of either m or n in the denominator rather than m−1 or n−1, respectively). We can diagonalise these “weighted” estimates of the covariance matrices. Those groups with larger sample sizes will have a commensurately larger influence on the covariance matrices (and means) than those groups with smaller sample sizes. This approach should therefore address problems in mean and covariance matrix estimation due to imbalances in sample sizes across groups. Indeed, this approach seems very similar (if not identical) to the weighting scheme proposed in [9]. Results from Experiment 2 from standard mPCA are denoted Experiment 2a below and results from the “weighted” covariance matrices Equations (7) and (8) are denoted Experiment 2b.

## 3. Results

Results for the eigenvalues from mPCA and single-level PCA for Experiment 1 are shown in Figure 3. The magnitude of these eigenvalues at level 1 via mPCA (with respect to the total variation) reduces strongly with increasing sample size per group $n_{l}$, as shown in Figure 3. The percentage variation at level 1 via mPCA also reduces strongly with increasing values of $n_{l}$, as shown in Table 2. Indeed, both measures are clearly tending towards the correct asymptotic value of zero in the limit of “infinite” sample size per group. The average sum of eigenvalues for single-level PCA over all MC simulations is (to within expected sampling error) equal to 300 for all values of sample size per group $n_{l}$. It is stated in [9] that mPCA underestimates the variation due to within-group effects (i.e., at level 2 of the mPCA model). However, we find that the sum of eigenvalues at level 2 for the mPCA model averaged over all MC simulations is also (again to within expected sampling error) equal to 300 in all simulations, which seems apparently to contradict this statement. Figure 4 shows that results for the sum of eigenvalues over both levels via mPCA extrapolate to the correct value of 300 in the limit $n_{l}\to \infty$. We see also from Figure 3 that the curves for the eigenvalues via single-level PCA and level 2 mPCA become flatter as we increase the sample size per group $n_{l}$, which agrees with the Marchenko–Pastur theorem [21].

Results for component scores are also given in Figure 3. As in [8], we find that strong apparent differences seem to occur between groups occurs at level 1 of the mPCA model. We see from Figure 3 that this occurs also via single-level PCA, albeit to a lesser extent. Again, these differences are due to random sampling effects and so are spurious. (Note that group centroids of component scores at level 2 via mPCA were indeed congruent with the origin for all MC simulated datasets and in all experiments carried out here). Differences between groups in Figure 3 become less pronounced for both mPCA and single-level PCA as the sample size per group is increased. Indeed, very strong overlap occurs in components scores and spurious differences between groups are quite small for a sample size per group of $n_{l}$ = 300 at level 1 via mPCA, as shown by the group centroids in this figure. Experiment 1 shows that random differences between groups that are spread over all 300 variables (and therefore probably also over possible principal components via traditional single-level PCA) are now being concentrated in just two components at the level 1 of the mPCA model. Experiment 1 indicates (as a “rule of thumb” only) that the sample sizes per group should at least be of similar magnitude to the number of variables, i.e., 300, in order to obtain reasonable results.

Results for the eigenvalues from mPCA and single-level PCA for Experiment 2a are shown in Figure 5. In this case, the sample sizes per group are varied for groups 1 and 2 only, whereas group 3 has $n_{3}$ = 10 in all simulations. The magnitude of these eigenvalues at level 1 mPCA shown in Figure 5 and percentage variation shown in Table 2 reduce with increasing sample size per group, *n*, although they are clearly “saturating” by *n* = 100 for groups 1 and 2. It is clear that the covariance matrices at both level 1 and 2 via mPCA are being very strongly affected by the small sample size in group 3. For example, eigenvalues at level 2 via mPCA exhibit a strange “spike” for low values of eigenvalue number. Eigenvalues at level 1 via mPCA are higher than those in Figure 3, where sample sizes are equal across all groups. However, we again find that the sum of eigenvalues for both single-level PCA and mPCA at level 2 is equal to 300 (again to within expected sampling error). Table 2 shows that the percentage variance does not tend to correct value of zero percent; a result is driven by the small sample size in group 3. Figure 4 shows that results for the sum of eigenvalues at both levels via mPCA do not extrapolate to the correct value of 300 in the limit $n_{l}\to \infty$ in Experiment 2a.

Component scores in Figure 5 show that group 3 is also a clear outlier for both mPCA, level 1 and single-level PCA. (Again, group centroids for mPCA at level 2 are congruent with the origin). It is noticeable that group 3 produces an outlying result in Figure 5 even for single-level PCA for $n_{1,2}$ = 300. Spurious differences are exaggerated for mPCA compared to single-level PCA, especially between groups 1 and 2 compared to group 3 for mPCA. Reasonable results for component scores via mPCA are never achieved in Experiment 2a due to the low sample size in group 3.

Results for the eigenvalues from mPCA and single-level PCA for Experiment 2b are shown in Figure 6. Again, the sample sizes per group are varied for groups 1 and 2 only, whereas group 3 has $n_{3}$ = 10 in all simulations. We see that eigenvalues for level 2 mPCA now are of very similar magnitude to results of single-level PCA. Indeed, we see that problems with leading eigenvalues for both levels 1 and 2 mPCA due to imbalances in sample sizes appear to have been removed in Figure 6 by the weighted form of the covariance matrices, which is an encouraging result. Eigenvalues for level 2, mPCA are very slightly lower in magnitude in Figure 6 than single-level PCA because Equations (7) and (8) are essentially population rather than sample covariance matrices, although this effect reduces quickly with increasing sample size per group $n_{1,2}$. Figure 4 shows that results for the sum of eigenvalues via mPCA extrapolate to the correct value of 300 in the limit $n_{l}\to \infty$ for Experiment 2b.

We see from Figure 6 that problems of spurious differences between groups are not removed by the weighted forms of Equations (7) and (8). Differences between groups that are contained in all variables (and again are probably spread over all components via single-level PCA) are again being concentrated in just two components at level 1 via mPCA. These spurious differences reduce strongly between groups 1 and 2 with increasing sample sizes in these groups, although they persist between groups 1 and 2 compared to group 3 even up to $n_{1,2}$ = 300, as shown in Figure 6. Reasonable results for component scores via mPCA are never achieved in Experiment 2b due to the low sample size in group 3, even when the weighted forms of the covariance matrices are used.

Figure 7 shows results for Experiment 3 in which between-group variation is added to the data, where the means of each group now follow a normal distribution with means equal to zero and a standard deviation of 0.25. Table 2 shows that mPCA is clearly tending towards the theoretical value of 5.9% with increasing sample size per group. Note that the sum of eigenvalues at level 2 via mPCA is again equal to 300 (within expected sampling error). Figure 4 demonstrates that the sum of eigenvalues over both levels via mPCA scale approximately linearly with $n_{l}^{-1}$ to a value of 318.55 in the limit $n_{l}\to \infty$. This is good agreement with the aysmptotic value of 318.75, although even better correspondence would presumably also be obtained by including higher values of $n_{l}$ in the regression data in Figure 4. Figure 4 shows also that the values for the sum of all eigenvalues via single-level PCA is approximately flat with respect to $n_{l}^{-1}$. Indeed, the total variation captured by single-level PCA is clearly well below the asymptotic overall total value of 318.75.

Figure 7 shows that both mPCA and single-level PCA overestimate differences between groups in component scores for small sample sizes per group. However, the broad pattern in the component scores for mPCA (and single-level PCA) has largely converged for sample size per group $n_{l}$ = 200 (not shown here) and it has certainly converged by the time that $n_{l}$ = 300 is reached (shown in Figure 7). Again, Experiment 3 indicates again (as a rough-and-ready “rule of thumb”) that reasonable results are obtained when the sample sizes in all groups are of similar magnitude to the number of variables, i.e., 300 here.

Results for the eigenvalues from mPCA and single-level PCA for Experiments 4 and 5 are shown in Figure 8 and Figure 9. These results for the correlated data in Experiments 4 and 5 are very similar to those earlier results from Experiments 1 to 3, which involved uncorrelated data. We see from Figure 8 and from Table 2 that variances at level 1 mPCA for Experiment 4 reduces with increasing sample size per group. Figure 10 shows that the sum of eigenvalues at both levels via mPCA scales approximately linearly with $n_{l}^{-1}$ for Experiment 4 and that this line extrapolates to a value that is very close that of single-level PCA in the limit $n_{l}\to \infty$. By contrast, Figure 9 shows that eigenvalues at level 1 do not tend to zero as the sample size per group increases for Experiment 5 and we note again that between-group variation has been explicitly added to the MC data in this case. Figure 10 shows that the sum of eigenvalues at both levels via mPCA scales approximately linearly with $n_{l}^{-1}$ for Experiment 5 and that it extrapolates in the limit $n_{l}\to \infty$ to a value that is much larger than that from single-level PCA. Table 2 shows that the percentage of variance explained by level 1 via mPCA for Experiment 5 converges to a non-zero value probably near to about 10%.

Component scores in Figure 8 and Figure 9 for Experiments 4 and 5 again also show a very similar pattern to those results in Experiments 1 to 3. Strong initial differences between groups in component scores via mPCA (and single-level PCA to some extent also) reduce strongly as sample sizes per group increased in Experiment 4. Indeed, it is noticeable in Figure 8 that differences between groups via mPCA are fairly small for $n_{l}$
*=* 100. By contrast, differences in component scores between groups via mPCA are observed for all sample sizes per group in Experiment 5, where between-group variation has been added explicitly to the MC data, for both single-level PCA and mPCA. Indeed, Figure 9 shows that differences between groups via mPCA are fairly similar for $n_{l}$ = 100 compared to $n_{l}$ = 300. Experiments 4 and 5 indicate again (and very broadly) that reasonable results are obtained when the sample sizes in all groups are of similar magnitude to the number of variables, i.e., 63 components for Experiments 4 and 5.

## 4. Discussion

An exploration of “pathologies” of mPCA was carried out here by considering a two-level mPCA model that is equivalent to bgPCA. It was clear that spurious differences between groups due to random sampling effects contained in all variables in Experiments 1, 2, and 4 were concentrated in the (relatively few) components at level 1 for mPCA. This effect meant that mPCA therefore falsely gave an impression of strong differences in component scores where in truth there were none. As stated in [8,9,17,18,19,20], pathologies of bgPCA (and therefore also mPCA) do exist, mostly strikingly in terms of interpretation of these component scores when sample sizes are low in any of the groups. However, these spurious differences in component scores via mPCA decreased significantly as the sample sizes per group were increased. 

Note that 3D facial scanning of human subjects can be costly because it can be time consuming and/or labour intensive. Sample sizes might often measured in tens of subjects only, where such “pathologies” are likely to manifest. However, sample sizes per group can be a problem even for large-scale epidemiological studies in humans if one is interested for example in subsets of subjects with rare syndromes that can affect facial appearance and shape. Similar problems might occur in archaeology or palaeontology, where the number of samples to be scanned might naturally be constrained. Another limitation of mPCA is that the rank of covariance matrices at higher levels of the model are limited to the number of groups minus one. In practice, this places a limit on the number of non-zero eigenvalues at these levels.

Imbalances in sample sizes in different groups can be addressed by using a form weighted covariance matrices inspired by the maximum likelihood solution, previously also suggested in [9]. Our results suggested that this “weighting” had a beneficial effect on covariance matrices and eigenvalues. However, weighting did not solve all of the problems of spurious differences in component scores between groups, which were due here to a very small sample size in one group. Experiment 2 demonstrates that misleading results for component scores persist via mPCA if the sample sizes are low in any of the groups. Notably, the usefulness of such “weighting” was questioned also in [8]. However, such weighting schemes might be useful when such imbalances occur and when sample sizes per group are sufficiently large enough in all groups. This topic requires more investigation in future.

Our calculations also indicated that single-level PCA underestimated the total amount of variance when between-group variation was introduced explicitly to the data generation model in Experiment 3. For example, Figure 4 showed that mPCA results extrapolated to a value that was very close to the theoretical asymptotic value for the total variation in Experiment 3, whereas single-level PCA did not. These results were also supported by evidence in Table 2. However, this is exactly what one would expect as the models used in MC data generation and via mPCA were essentially identical. Very similar results were seen in Experiment 5 where between-group variation was introduced to correlated 3D MC data representing 3D facial shape. Traditional PCA is essentially just a single-level method and so one would not expect it to capture the effects of such multilevel structures and/or of “clustering.” Interestingly, the sums of eigenvalues via mPCA scaled approximately linearly with the inverse of the sample size per group in all Experiments 1 to 5, which is another potentially important result of this research. We speculate that this might be another manifestation of the Marchenko–Pastur theorem [21].

The simulations presented here for the uncorrelated normally distributed variables in Experiments 1 to 3 are the most severe (and artificial) test of both mPCA and single-level PCA as any apparent structure to the data is due purely to random sampling effects. It was noted (e.g., in [9]) that these problems of spurious differences between group for bgPCA are reduced when the multivariate data is correlated, which generally is the case in reality, e.g., for shape data. The evidence from Experiments 4 and 5, in which correlated multivariate normally distributed data was generated, were inconclusive in relation to this claim specifically, although they do not contradict it. However, the total number of variables was much lower in Experiments 4 and 5 compared to Experiments 1 to 3, which makes it harder to compare results on an equal footing. Experiments 4 and 5 do underline that the results presented here are clearly relevant to modelling shape, such as those illustrated by Figure 1 for 3D facial shape and as described in [11,12,13,14,15,16,17].

The results of this work show broadly that reasonable results ought to be obtained when sufficiently large sample sizes per group are used in all groups. As a “rule of thumb” only, sample sizes per group in all groups should be at least equal to the number of variables. However, modes of variation from mPCA should also always be examined critically and they should be compared to known results in the literature where they are known to exist, e.g., known changes in facial shapes in humans due to sex [12]. The author of [8] presents a detailed list of recommendations about the use (or refraining from use) of bgPCA in relation to biological morphometrics. The authors of [19] propose cross-validation as a method of overcoming these problems, whereas the authors of [20] propose a mixture of permutation tests and cross-validation. The interested reader is referred to [8,19,20] for more details.

## Figures and Tables

**Figure 1 jimaging-08-00063-f001:**
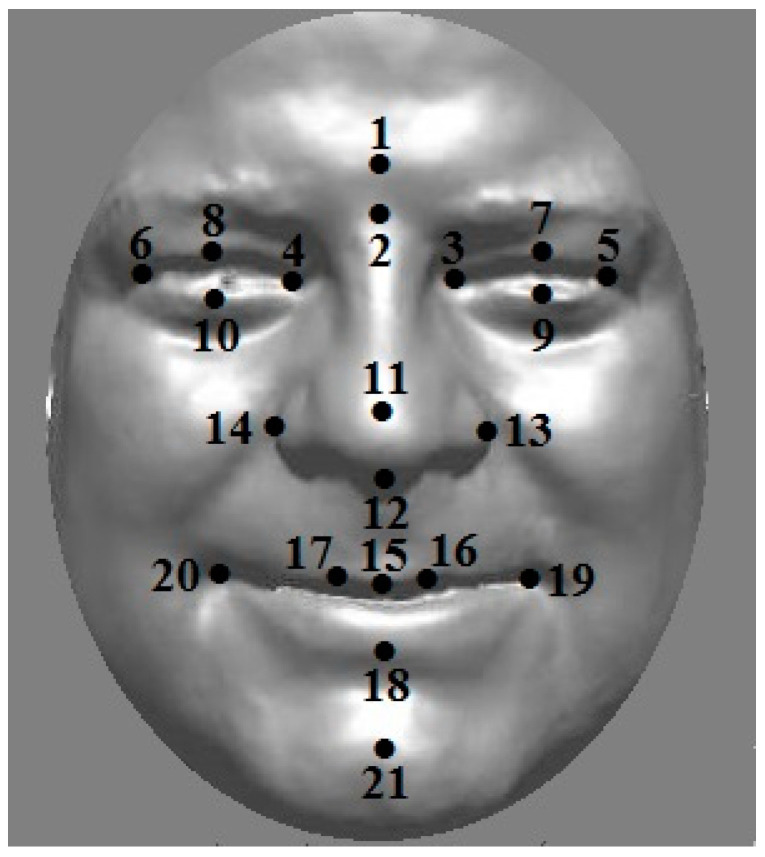
Twenty-one anthropometric landmarks placed on a 3D scan of the author’s face.

**Figure 2 jimaging-08-00063-f002:**
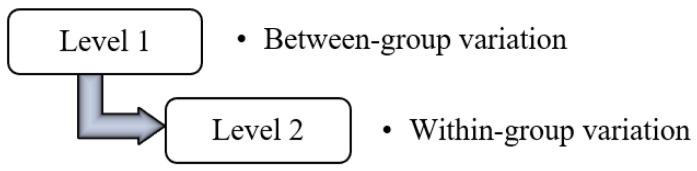
Schematic of a two-level multilevel model used here in mPCA calculations.

**Figure 3 jimaging-08-00063-f003:**
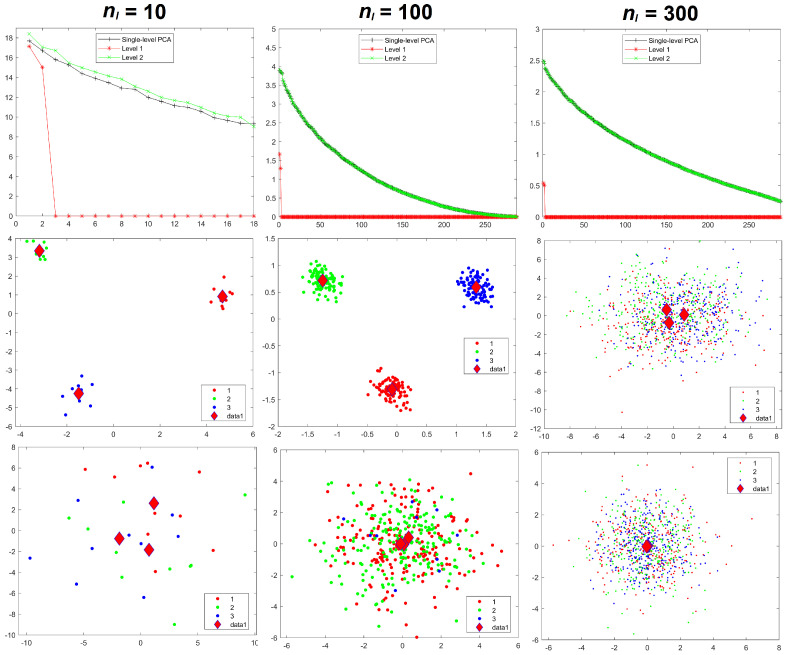
Experiment 1: eigenvalues (**upper row**), mPCA, level 1 component scores (**middle row**), and single-level PCA, component scores (**bottom row**) for sample sizes per group of $n_{l}$ = 10 (**left-hand column**), $n_{l}$ = 100 (**middle column**), and $n_{l}$ = 300 (**right-hand column**) in all groups l=1,2,3. Group centroids for component scores are shown by the diamonds; *x*-axis = first component; *y*-axis = second component.

**Figure 4 jimaging-08-00063-f004:**
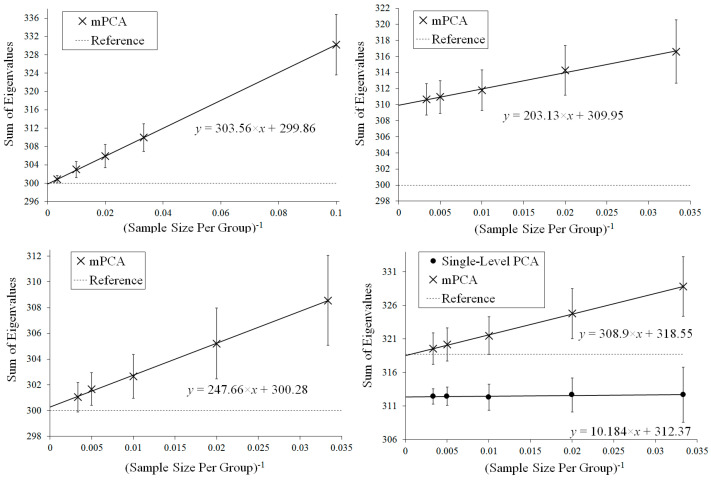
Extrapolation of the mean (over all MC simulations) sum of all eigenvalues for mPCA in the limit sample size per group $n_{l}\to \infty$ for Experiment 1 (**top left**), Experiment 2a (**top right**), Experiment 2b (**bottom left**), and Experiment 3 (**bottom right**). These values via mPCA scale approximately linearly with $n_{l}^{-1}$. Reference values for the asymptotic estimates of the total variance are shown by the dashed lines in these figures (equal to 300 for Experiments 1 and 2 and to 318.75 for Experiment 3). Results of single-level PCA for Experiment 3 are approximately “flat” with respect to $n_{l}^{-1}$. (Standard errors are shown by the error bars).

**Figure 5 jimaging-08-00063-f005:**
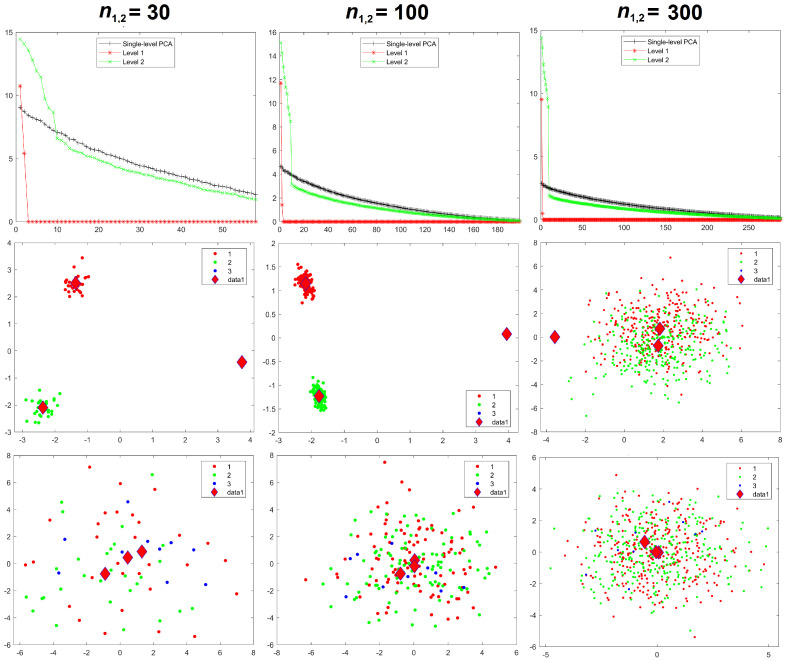
Experiment 2a: eigenvalues (**upper row**), mPCA, level 1 component scores (**middle row**), and single-level PCA, component scores (**bottom row**) for sample sizes per group of $n_{1,2}$ = 30 (**left-hand column**), $n_{1,2}$ = 100 (**middle column**), and $n_{1,2}$ = 300 (**right-hand column**) in groups 1 and 2. Note that $n_{3}$ = 10 in group 3 in all simulations for Experiment 2a. Group centroids for are shown by the diamonds; *x*-axis = first component; *y*-axis = second component.

**Figure 6 jimaging-08-00063-f006:**
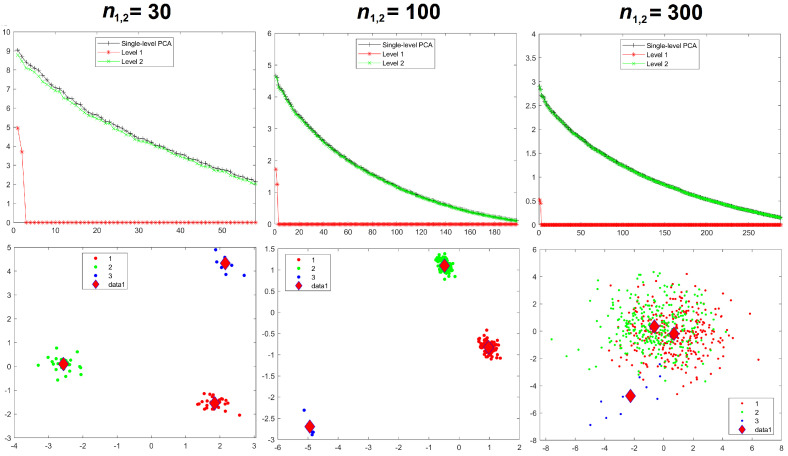
Experiment 2b: eigenvalues (**upper row**) and mPCA, level 1 component scores (**bottom row**) for sample sizes per group of $n_{1,2}$ = 30 (**left-hand column**), $n_{1,2}$ = 100 (**middle column**), and $n_{1,2}$ = 300 (**right-hand column**) in groups 1 and 2. Note that $n_{3}$ = 10 in group 3 in all simulations for Experiment 2b. Group centroids are shown by the diamonds; *x*-axis = first component; *y*-axis = second component. (Results for single-level PCA are as shown in Figure 5).

**Figure 7 jimaging-08-00063-f007:**
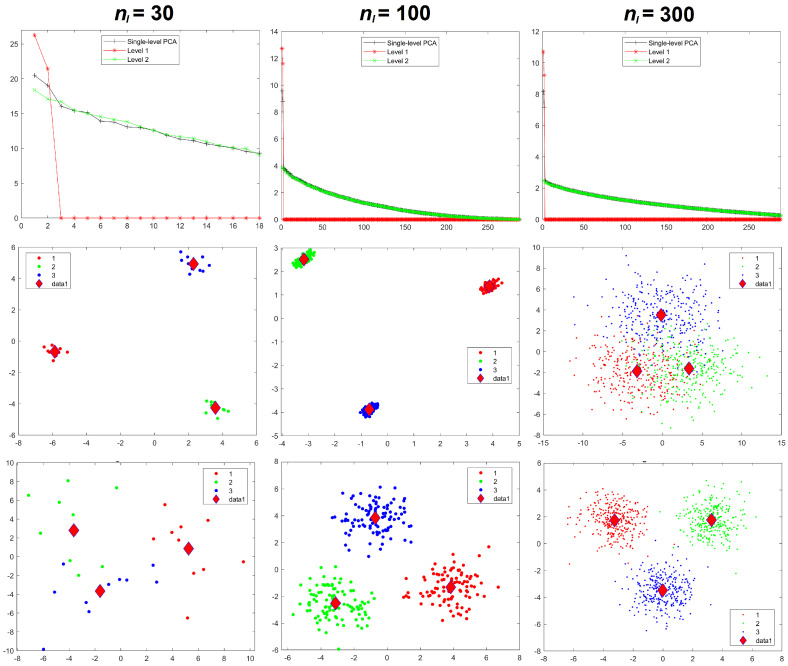
Experiment 3: eigenvalues (**upper row**), mPCA, level 1 component scores (**middle row**), and single-level PCA, component scores (**bottom row**) for sample sizes per group of $n_{l}$ = 10 (**left-hand column**), $n_{l}$ = 100 (**middle column**), and $n_{l}$ = 300 (**right-hand column**) in all groups l=1,2,3. Between-group variation has been added in this case and so component scores should be strongly separated for all values of $n_{l}$. Group centroids are shown by the diamonds; *x*-axis = first component; *y*-axis = second component.

**Figure 8 jimaging-08-00063-f008:**
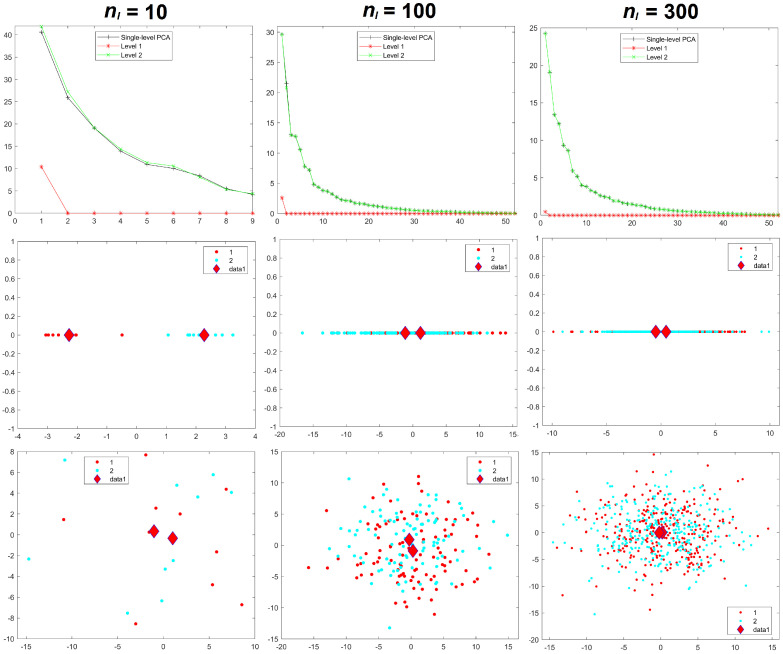
Experiment 4: eigenvalues (**upper row**), mPCA, level 1 component scores (**middle row**), and single-level PCA, component scores (**bottom row**) for sample sizes per group of $n_{l}$ = 10 (**left-hand column**), $n_{l}$ = 100 (**middle column**), and $n_{l}$ = 300 (**right-hand column**) in both groups l=1,2. Group centroids are shown by the diamonds; *x*-axis = first component; *y*-axis = second component.

**Figure 9 jimaging-08-00063-f009:**
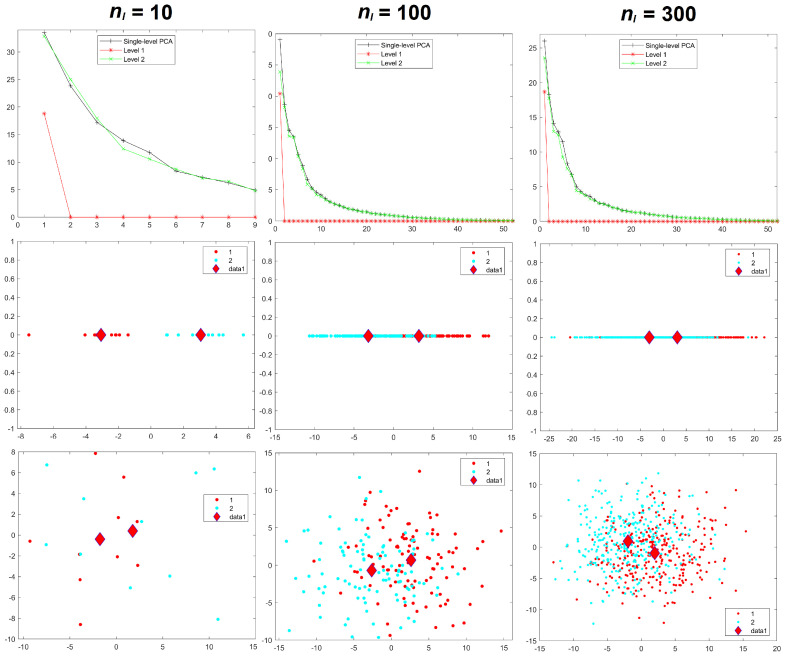
Experiment 5: eigenvalues (**upper row**), mPCA, level 1 component scores (**middle row**), and single-level PCA, component scores (**bottom row**) for sample sizes per group of $n_{l}$ = 10 (**left-hand column**), $n_{l}$ = 100 (**middle column**), and $n_{l}$ = 300 (**right-hand column**) in both groups l=1,2. Group centroids are shown by the diamonds; *x*-axis = first component; *y*-axis = second component.

**Figure 10 jimaging-08-00063-f010:**
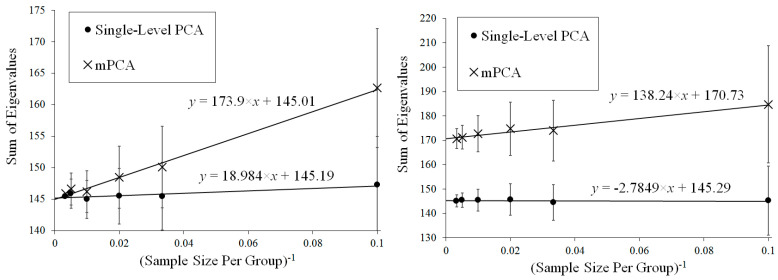
Extrapolation of the mean (over all MC simulations) sum of all eigenvalues for PCA and mPCA in the limit sample size per group $n_{l}\to \infty$ for Experiments 4 (**left**) and 5 (**right**). The values for mPCA again scale approximately linearly with $n_{l}^{-1}$. Results of single-level PCA are approximately “flat” with respect to $n_{l}^{-1}$. (Standard errors are shown by the error bars).

**Table 1 jimaging-08-00063-t001:** Overview of all MC simulations carried out here. Experiments 1 to 3 use uncorrelated normally distributed variables, whereas Experiments 4 and 5 use correlated normally distributed data inspired by 21 3D landmark points (thus 63 variables), as shown in Figure 1. (# Variables = number of variables; Correlated? = whether or not these variables are correlated or uncorrelated; BG Variation? = whether or not between-group variation is used in data generation; Balanced? = whether or not sample sizes are equal in all groups; Weighted? = whether or not weighted covariance matrices of Equations (7) and (8) are used).

	Exp. 1	Exp. 2a	Exp. 2b	Exp. 3	Exp. 4	Exp. 5
# Variables	300	300	300	300	63 (=3 × 21)	63 (=3 × 21)
Correlated?	No	No	No	No	Yes	Yes
BG Variation?	No	No	No	Yes	No	Yes
Balanced?	Yes	No	No	Yes	Yes	Yes
Weighted?	No	No	Yes	No	No	No

**Table 2 jimaging-08-00063-t002:** Mean (over all MC simulations) of percentage variance of Equation (3) explained by level 1 via mPCA for Experiments 1 to 5. Experiment 2a using standard mPCA, whereas Experiment 2b uses the weighted “population” covariance matrices of Equations (7) and (8). Reference values are given via asymptotic estimates for Experiments 1 to 4 and from experimental data for Experiment 5 ($n_{1}=124;\;n_{2}=126$). (Standard errors are shown in brackets).

	Exp. 1 $\left(n_{3}=n_{1,2}\right)$	Exp. 2a $\left(n_{3}=10\right)$	Exp. 2b $\left(n_{3}=10\right)$	Exp. 3 $\left(n_{3}=n_{1,2}\right)$	Exp. 4 $\left(n_{1}=n_{2}\right)$	Exp. 5 $\left(n_{1}=n_{2}\right)$
$n_{1,2}$ = 10	11.5% (0.63%)	11.5% (0.63%)	8.7% (0.51%)	17.4% (2.20%)	10.2% (4.85%)	19.2% (5.01%)
$n_{1,2}$ = 30	3.3% (0.16%)	5.4% (0.35%)	3.1% (0.17%)	9.0% (0.50%)	3.1% (1.20%)	12.8% (2.59%)
$n_{1,2}$ = 50	2.0% (0.11%)	4.5% (0.27%)	1.9% (0.12%)	7.7% (0.41%)	2.0% (0.84%)	12.2% (2.31%)
$n_{1,2}$ = 100	1.0% (0.06%)	3.9% (0.25%)	1.0% (0.06%)	6.7% (0.39%)	0.9% (0.33%)	11.3% (1.55%)
$n_{1,2}$ = 200	0.5% (0.03%)	3.6% (0.26%)	0.5% (0.03%)	6.3% (0.33%)	0.5% (0.21%)	10.7% (1.03%)
$n_{1,2}$ = 300	0.3% (0.02%)	3.5% (0.30%)	0.3% (0.02%)	6.1% (0.35%)	0.3% (0.12%)	10.6% (0.87%)
Reference	0%	0%	0%	5.9%	0.0%	10.4%

## Data Availability

Synthetic data was generated using Monte Carlo methods specified in the Section 2.1 and so this is not applicable.

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
