# Peer review of "An Exploration of Pathologies of Multilevel Principal Components Analysis in Statistical Models of Shape"

_2313-433X, 2022, doi:10.3390/jimaging8030063_

Round 1

Reviewer 1 Report

In general and timely paper. Confirms prior work by others. On page 2 line 46 it will be helpful to readers to also cite two additional papers that are follow-ups to the citations:

Cardini, A. and P. D. Polly (2020). "Cross-validated Between Group PCA Scatterplots: A Solution to Spurious Group Separation?" Evolutionary Biology 47: ?85-95.

Rohlf, F. J. (2021). "Why Clusters and Other Patterns Can Seem to be Found in Analyses of High-Dimensional Data." Evolutionary Biology 48(1): 1-16.

Line 46: it might also be useful to mention that it also relates to canonical variance analysis.

Line 58: reference number  8 does not discuss the effect of having unbalanced sample sizes.
Him him him him him him

Page 3 line 71: it should be noted here that this is the place absolute limits on the magnitudes of the eigenvalues. It only shows what is expected for random data. Clearly, one could construct perfectly correlated data such that the first eigenvalues will be large and the rest zero.,

Page 3 line 92:  what does it mean to  "standardize the length scales"? There is a step two reduce to centroid size 1. Is that what is meant?

Page 3, 109: is there a reference term "multilevel principal components analysis"? It seem to just performing a PCA at any level of a MANOVA.  While one could could obviously do that, it would usually be of less interest because one typically has so few degrees of freedom at the highest levels.

Page 4, section 3.3: I am not sure of the need for giving the equations here for a MANOVA. Not new.

Page 5: Why does section 3 come after section 3.3?

Page 5, figure 3: most of the text in the figures is much too small.

Page 6, line 183: Better to day "within expected sampling error" than "statistical accuracy". Also on lines 288-9.

Page 7, line 228: Yes, as expected and shown in [9].

Page 13, line 394: Agreed. However in many current studies (e.g. primate evolution using fossils) it is just not possible to obtain large sample sizes.

Author Response

Responses to Referee 1 are given in the attached file.

Reviewer 2 Report

Some issues I detected during reading the paper:

  1. Abstract. '… we confirmed previous results of other researchers...' To my knowledge, the articles 8 and 9 do not use MC simulations, so I would highlight this in the abstract and in the article.
  2. 2.1. I would appreciate to summarize simulation design in a table which would prevent the reader from searching for details later in the results and give a clear overview about simulation study. It wouldalso be better to state all sample sizes used in the text here. Is 300 defined as 3x100 landmarks? It is not clear.
  3. 2.1. I am wondering if the author used Procrustes superimposition in the simulation study. Normal variables must be submitted to either Procrustes superimposition with fixed (equal for all individuals) or optimized centroid size. If not, there is an artificial deformation present in the data and could affect the results of the simulation study.
  4. 2.2. In Eq. (1), I presume that the index 'i' in both mu is redundant.
  5. 2 and 3. In summation indices, sometimes 'i' is used only and in other cases 'i = 1” (index “i” is chosen here arbitrarily). I would be best to use the latter in the entire paper.
  6. Simulation study design. I would appreciate at least some discussion consistent with the trends in the results about sample size higher than number of landmarks.
  7. In many practical situations, we have not only landmarks but also curves and surface patches with many semi-landmarks and then the sample size might be much smaller than number of dimensions. So we almost never reach number of semi-landmarks times three being smaller than sample size, i.e. how far we can go with a difference between dimension and sample size? The author should add some discussion about this problem.
  8. The figures should be exported in vector graphics and not as raster, i.e., PDF would be preferable for better image quality in electronic version of the paper.

Author Response

Responses to Referee 2 are given in the attached file.
